# Coproduced, arts interventions for nurturing care (0–5 years) in low-income and middle-income countries (LMICs): a realist review

Nicola Kay Gale ,[1] Kalim Ahmed,[1] Niélé Hawa Diarra,[2]
Semira Manaseki-Holland,[1] Evans Asamane,[1] Cheick Sidya Sidibé ,[2]
Ousmane Touré,[2] Michael Wilson,[3] Paula Griffiths[3]

¹University of Birmingham, Birmingham, UK
²University of Bamako, Bamako, Mali
³Loughborough University, Loughborough, UK

**Correspondence to**
Dr Nicola Kay Gale;
n.gale@bham.ac.uk

## ABSTRACT

**Objectives** Community-based arts interventions have the potential to support contextually relevant nurturing care programmes and policies that adapt to different settings. Understanding the distinctive features of using the arts in local, culturally specific ways in low/middle-income countries (LMICs); how this varies by context; and gaining a better understanding of the perspectives on desirable outcomes for communities is important evidence that this review generates.

**Design** We conducted a realist review of papers that covered outcomes related to child health or development (0–5 years) AND arts-based approaches AND community-based, participatory approaches AND based in LMICs using a range of databases and other networks. A coding framework was developed covering context, intervention, outcomes, mechanisms, study, sustainability, transferability and scalability.

**Results** The included papers reported 18 unique interventions. Interventions covered 14 countries, with evidence lacking for South America, Arab countries and parts of Africa. Lead authors came from mostly clinical science-based disciplines and from institutions in a different country to the country/countries studied. Intended outcomes from interventions included clinical, health systems/organisation, changes in practices/behaviours/knowledge/attitudes, and wider social and educational goals. We identified three demi-regularities (semi-predictable patterns or pathways of programme functioning): participatory design based on valuing different sources of expertise; dynamic adaptation of intervention to context; and community participation in arts-based approaches.

**Conclusions** Our findings suggest that arts-based, nurturing care interventions have greater potential when they include local knowledge, embed into existing infrastructures and there is a clear plan for ongoing resourcing of the intervention. Studies with better documentation of the lessons learnt, regarding the intervention delivery process and the power dynamics involved, are needed to better understand what works, for whom and in which contexts.

## BACKGROUND

This article reports the findings from a realist review and synthesis of the published literature on community-based arts interventions targeted at improving health and development of 0–5 years old in low/middle-income countries (LMICs).

### Nurturing care

UNICEF's nurturing care framework provides a structure through which countries or communities can support parents and caregivers to provide an environment that ensures optimal health, nutrition and development for infants and young children and in which these individuals can live free from threats and have opportunities to develop and learn using responsive interaction.[1] More than 5 million children did not survive to their fifth birthday in 2020.[2] Furthermore, 43% (over 250 million) infants and young children in LMICs do not achieve their developmental potential for reasons including poverty, undernutrition, and a lack of appropriate, responsive and nurturing care and stimulation.[3 4] Appropriate nurturing care has the potential to reduce mortality and morbidity and improve early childhood development.[5] To support the core elements of the nurturing care framework, programmes and policies are needed that are contextually

---

### STRENGTHS AND LIMITATIONS OF THIS STUDY

⇒ Systematic search of the international literature ensured good coverage of the field.
⇒ Multi-disciplinary team authorship ensured realist synthesis methods could capture what works, for whom and in what circumstances.
⇒ Searches were conducted in English so the review may have regional biases, as well as positive results publication bias.
⇒ Inclusion of studies that report interventions meant that relevant contextual or conceptual papers, books or chapters, more common in the arts disciplines were not included.

relevant and take account of the requirements and different needs of all communities, because nurturing care is heavily embedded in local cultures and those seeking to implement health or child development interventions are influenced by local political and economic circumstances.[6 7]

## Power and participation

Community-based interventions are often coproduced with local communities. The term coproduction is highly debated, but broadly refers to a collaborative and reciprocal process of exchange between actors of differing backgrounds, to generate outcomes uniquely embedded in the different perspectives shared.[8] There are often significant challenges in implementing coproduction in practice, such as knowledge and power asymmetries between those collaborating. Often power dynamics in these collaborations shift away from local expertise to professional expertise meaning that not all voices are heard.[9–14] Such power dynamics may produce interventions which are not sensitive to the specific issues, barriers and affordances of the context in which the intervention is implemented.

Many have argued that to overcome challenges and realise the purported benefits of coproduction, collaborators must move beyond a 'tick box' or process-driven approach to involvement towards partnerships that enable partnership synergy.[11] This is built through mutual trust and understanding between parties and takes into account contextual factors.[15 16] Careful consideration of how interventions may draw on and influence local infrastructures and relationships is also important, to build sustainable collaborations.[17 18]

## Arts-based approaches

Applied arts is an umbrella term that refers to work that seeks to use the arts as a tool to be applied to a social issue; other commonly used terms in the literature are Arts for Development, Arts for Social Change or Participatory Arts.[19 20] Arts-based interventions are usually carried out in group settings where they are used to undertake collective processing and/or for the communication of new information such as health advice, for example, through dramatic or musical performances, bazaar event days with activities, competitions and prizes. Applied arts approaches are distinct from therapeutic approaches, such as Art Therapy, which are associated with improving individual health directly.[21]

When practiced through local arts traditions, an intervention is embedded into local culture rather than imposed from outside, which is typically thought to make the intervention more relevant, coherent and engaging to community members including using existing community strengths.[22] When both performer and audience are involved in a live performance it can act to bind communities with a sense of togetherness, increased trust and common goals.[22 23]

## Rationale for review and objectives

Given the growing global interest in arts-based approaches to health and child development interventions, there is a clear need to understand whether they are effective and, if so, how they work. There is a growing understanding of the effectiveness of arts-based approaches,[23–30] and much scholarship in this field from the arts disciplines engages with issues such the artistry or process of artistic performance or creativity, or the role of artistic practice in interrogating concepts or cultures,[31 32] but arts scholarship tends to focus less on evaluating interventions per se. Systematic reviews of the effectiveness of arts-based interventions in child health and development in LMICs are lacking.

Realist evaluation methodologies aim to ensure that context is incorporated in any analysis of effectiveness, using the refrain, 'what works, for whom and in what circumstances?'.[33–35] Interventions that seek to address issues of improving nurturing care are often complex, addressing many layers of social and cultural practice, and influenced by political and economic context, so oversimplistic assessments that deem such interventions categorically 'effective' or not are problematic. The nuanced realist evaluation of arts-based nurturing care interventions in LMICs provided in this review will provide evidence to inform policy and intervention design to improve nurturing care in some of the world's most vulnerable contexts. It will also explore the sustainability of interventions and generalisability of findings, and specifically whether interventions might be reproducible in other contexts (usually with some adaptations to context), or scalable to a wider population.

To address these gaps, we conducted the first realist review and synthesis of the published literature on community-based arts interventions targeted at improving health and development for 0–5 years old in LMICs. We identify (i) possible intervention mechanisms focusing on the distinctive feature of using the arts in local, culturally specific ways; (ii) relevant issues of contextual difference and (iii) a broad perspective on desirable outcomes for communities, rather than focusing solely on medical outcomes. Realist review is an effective method for analysing complex interventions that are highly context-dependent.[33 34] This can help inform future research as well as policy and intervention design and implementation in this field, by identifying what works, for whom and in what circumstances, and what elements are likely to make interventions more sustainable, transferable or scalable. Additional resources about realist reviews can be found the RAMESES project website.

The overarching review questions are:
► What kinds of coproduced, arts-based, nurturing care interventions have worked in LMICs, for whom and in what circumstances?
► To what extent are successful interventions sustainable, transferable or scalable?

## METHODS

We applied the RAMESES quality and reporting guidelines.[35]

### Scoping the literature

We worked with a team in the UK and Mali, including the intended users of the review, to clarify focus and prioritise research questions. We presented our initial ideas and later our preliminary findings to the MaaCiwara project steering committee. MaaCiwara is an ongoing study with collaborations between institutions in Mali and the UK examining the impact of an intervention in Mali for nurturing care, with significant arts-based elements.[36] Our preliminary objectives for this review were to identify examples of arts-based interventions coproduced between communities affected by poor child health, nutrition, and development outcomes and those involved in the public health/health promotion sector in LMICs and/or research organisations; to develop an understanding of the theoretical frameworks that inform these interventions or can help explain their effects, and to understand the mechanisms that contribute to success for these interventions. As the review progressed, we introduced a second research question focused on the long-term potential of these sorts of interventions to have an impact within the original context and in other similar ones. The findings from this review have been used to inform the data collection and analysis strategy for the primary empirical research.

### Search strategy

We searched for peer-reviewed journal articles that covered outcomes related to child health or development (0–5 years) AND arts-based approaches AND community-based, participatory approaches AND based in LMICs (online supplemental table 1). An experienced information specialist performed these searches. These terms were used to search EBSCO, OVID, ProQuest, Scopus, Web of Science and Child Adolescence and Education Studies databases.

### Selection and appraisal of studies

KA and NHD independently screened titles and abstracts for relevance. Where there was uncertainty, further discussions took place with PG and NKG and we refined our relevance criteria in line with the outcomes of these discussions (online supplemental table 2). The gradual refinement of these criteria helped focus the review questions and ensured coherent conceptualisations of the research area between research team members.

### Data extraction

Given the wide range of intended outcomes and evaluation methods used, we did not attempt to classify in any categorical way whether or not an intervention had been effective, rather we attempted to unravel whether the papers were able to say something about 'what works, for whom and in what circumstances' and whether improvements could be sustained. We did

this by identifying and analysing context-mechanism-outcome configurations from information that was available in the papers.

To address the first research question, the team developed an initial coding framework with six *categories* of codes:

► *Context*: country-region, socio-cultural, economic-developmental, legal-political, geographical-physical.
► *Intervention*: agents, coproduction, arts-based methods, dates, design, implementation.
► *Outcomes*: impact on primary outcome; impact on secondary outcomes, unintended outcomes/ consequences.
► *Mechanisms*: observed-evidenced; theorised.
► *Study*: aims, constructs-variables, date, limitations, sampling, theoretical framework, study design/ methods, first author location.
► *Paths*: interaction between context/intervention, context/outcomes, intervention/outcomes and context+intervention→outcomes.

The 'paths' category of codes was based on anything in the paper that spoke direct to the issues of the context-mechanism-outcome configuration,[33] even if a realist approach was not used. After agreeing the approach, papers were coded and data were extracted independently by KA and NHD, who then had a series of discussions to resolve differences.

To address the second review question, a second round of data extraction was conducted by NKG and PG on reported or potential issues of:

► Sustainability
► Transferability
► Scalability

### Data analysis and synthesis

Our process for generating theories and adjudicating between them involved four stages involving NKG, KA and PG meeting regularly, with emerging themes discussed in wider team meetings. First, we worked to compare and contrast the different interventions and the evaluation methods used, with specific reference to the ways in which the interventions focused on the key concepts of our research question, such as 'coproduction' and 'community arts-based'. Second, we looked for demi-regularities in the data[37] in order to generate a set of propositions about the social mechanisms at play in these interventions. Our discussions cohered around three themes for further exploration: developing sustainable partnerships; tailoring interventions to local context, and using arts-based approaches that engage with social norms. Third, we appraised each intervention with a series of questions centred on these three themes (online supplemental table 3), in order to answer the question 'what works, for whom and in what circumstances?'. Finally, we considered whether the interventions were reported to be or likely to be sustainable, transferable and/or scalable.

**Patient and public involvement**

No patients or members of the public were involved in this literature review.

## RESULTS

### Study characteristics

Our database searches generated 940 unique results (online supplemental table 4). After an initial screening of titles and abstracts, 175 results remained. At full text screening, 16 were deemed relevant to the review. Additionally, seven papers from our informal search and consultation with the wider reference group were identified which met these inclusion and exclusion criteria and were added to our sample. Therefore, 23 papers were included in the review, which reported findings from 18 separate interventions (process overview in online supplemental figure 1). In some cases, formative research or baseline characteristics were covered separately from evaluations of the effectiveness of the interventions resulting in multiple papers per intervention. The 18 interventions are summarised in online supplemental table 5.

### Global coverage

These interventions covered 14 different countries, with the majority focussing on the Asia Pacific region (Bangladesh (n=3), Cambodia (n=2), Haiti, India, Malaysia, Nepal (n=2) and Vietnam), followed by the African region (Ethiopia, Kenya, The Gambia, Nigeria (n=2), Uganda and Zambia (n=2)), and one paper from South America (Peru). This shows that the English language evidence base is not providing much evidence for the South American or Arab regions, as well as large parts of Africa.

### Authorship location and disciplines

Given the focus on coproduction in the review and the interdisciplinary nature of the interventions, there was a notable imbalance in the pattern of authorship. It was often unclear whether the projects had been initiated by those leading the publication or by others, such as local teams, but out of 23 papers, 17 have first authors affiliated to an institution in a different country to the country or countries being studied in the paper. In addition, most academic outputs were led from the health sciences and only one paper was led by researchers in the arts and humanities, two by social scientists, and one by a team in engineering.

### Intended outcomes

There was a range of different intended outcomes from interventions, from the clinical (eg, maternal mortality rates, malaria rates, anthropometry), the organisational (eg, utilisation of services), to changes in practices/behaviours (eg, changed dietary practices, exclusive breast feeding, sharing of messages within families), and changes in knowledge or attitudes (eg, dissemination of messages, health information retention) and wider social and educational goals (eg, women's empowerment, school readiness).

### Evaluation methods

A range of qualitative and quantitative methods were used to evaluate the interventions, including cluster randomised controlled trials or impact assessments (non-blinded), quasi-experimental studies, a prospective cohort study, pre-post outcome evaluations (with or without control), cross-sectional surveys, interviews, focus groups and digital story creation. In a few cases, the focus was on descriptive case studies or process descriptions, rather than on evaluating outcomes.

### Main findings

We identified three main demi-regularities: participatory design based on valuing different sources of expertise; dynamic adaptation of intervention to context; and community participation in arts-based approaches. These three pathways were unevenly realised across the different interventions described and evaluated in the papers (online supplemental table 6) but can serve as a basis for future studies, or for those designing new or adapting interventions to reflect on.

### Programme theory 1: participatory design based on valuing different sources of expertise

While our inclusion criteria required some element of 'coproduction', there was a huge range of approaches, including those that were clearly researcher-led, with limited involvement of local people, local artists or other stakeholders, those that were driven by the interests of funders, particularly NGOs, and those where local people, local artists or local researchers drove the projects forward. In some cases, the coproduction elements were primarily enacted during the design stages,[38] while others were mainly at the implementation stage through the adaptation of an existing intervention to local context.[39] Others provided continuing opportunities for communities to engage throughout the length of the programme.[40]

Coproduction models ranged from very light touch coproduction models to coproduction that started at the beginning of the project and followed throughout the programme of activities. At the lighter end was a model where community health volunteers talked with local communities in discussion groups to get their involvement in the intervention, but where the intervention was designed externally to the community.[41] Other lighter touch coproduction models included a programme of activities developed outside of the community but delivered with local artists/drama groups to engage communities in the production of the intervention success[42] or a model where an existing intervention was adapted with local community input.[43] At the other end of the spectrum, some projects used coproduction models that started from the beginning of the project to design the intervention in partnership with the community,[38] or where communities develop the materials and stories

that drive the intervention's focus messages,[44] or where the intervention was framed around collective learning between artist and the community.[40]

There were some limitations in the reporting of coproduction approaches. Often there was very little critical reflexive elaboration about the power structures in which the interventions were being designed and implemented, or what the challenges and benefits of harnessing community power structures were (such as who was included and who was not in the process). Another area that was underdeveloped was to what extent the resourcing of the intervention or the evaluation influenced the coproduction. It is likely that there were pragmatic requirements to deliver public health interventions for funders, combined with potent power dynamics in communities, which could intensify the exclusion of already 'hard to reach' groups.

## Programme theory 2: tailoring of intervention to context

All papers recognised to some degree the need to develop interventions what were tailored to the local context and employed a range of different methods to do so. These ranged from formal formative research processes that used literature reviewing, as well as qualitative (interviews, visual methods) and quantitative methods (surveys, analysis of routine or existing data sets, media audits) to understand the pre-existing social norms and behavioural patterns related to nurturing care,[43 45 46] or to identify key messages/changes to focus on. In other studies, a pilot or series of pilots were used to refine the intervention, drawing on feedback from those delivering, receiving, and funding the interventions.[38]

Some interventions were built around existing community groups, such as 'Fathers Clubs' in rural Haiti, where the social groupings were already well established and health messages were incorporated into these in a way that was led by the fathers themselves, with support from health workers,[47] or the women's groups in rural Zambia.[45] In addition, existing infrastructure was often used, such as health facilities or community meeting spaces.[41 48]

Some papers acknowledged the importance of enabling or constraining environments, both physical (such as the location of handwashing stations or providing spaces for infant feeding) and not just focusing on health education messages. The importance of the social environment (such as the need to change perceptions about using soap in mass media campaigns)[49] was also identified. These approaches supplemented or complemented the arts components.

A significant limitation in the data was that authors did not always report on the opportunities for feedback from stakeholders, and so it is unclear whether those opportunities existed or not. Even where feedback opportunities are mentioned, it is not always clear what impact the feedback had on the subsequent delivery of the intervention, if any. Where performing arts were part of the intervention, there was very little reported in papers on how performing arts practitioners contributed to the overall design or content of the performance, so it is hard to judge the extent to which they influenced the artistic approaches taken.

Potentially relevant to the context was other similar or related public health or NGO programmes that were being undertaken in the region, and any impact they could have on outcomes, but these were rarely reported.

## Programme theory 3: community participation in arts-based approaches

Types of arts-based activities were wide-ranging and included: storytelling, story-acting, visual cues, interactive role-play, posters, comic books, nursery rhymes and songs, puppet shows, drama, drawing, singing competitions, animations, stories, TV promotions, games, dancing, fashion shows, comedy sketches, speeches, professional singing performances, street drama, skits (sketches), kitchen makeovers, decorating kitchens, letter exchanges, family drama, folk songs, cookery demonstrations, photo displays, Kalajarta (folk theatre), rupakas (musical dramas), writing scripts, community videos, TV spots with mini-dramas of intervention messages, radio messaging, brochure design, digital storytelling, drumming, testimonials, cartoon films, mass media campaigns, game based and music based education, recitals, storybook reading, interactive games and watching videos.

Less than half of the interventions delivered the arts components in a way that included specialist arts practitioners, such as local drama groups, traditional communicators, traditional theatre groups, songwriters, photographers and other artists. Most interventions delivered the arts components through or with other members of the community, such as community health workers (normally with basic education), parents (mostly mothers) and wider community groups. In some cases, workers from NGOs were used to deliver intervention materials. It was much less common for papers to mention local or national government agents as involved in intervention delivery.

Very few papers commented on their theories or principles of how the arts were expected to support the intervention, with the exception of Olaide's paper (written by a Performing Arts scholar) which noted that it drew on Boal's concept of 'forum theatre' which is part of a broader concept of 'theatre of the oppressed', which is a widely used tool particularly in India, and Eastern and Southern Africa to facilitate social change.[50] Most studies were reported by health scientists and in health journals so in very few cases was a detailed understanding of the process of arts production or performance interrogated in a critical or scholarly way. The intended mechanisms of change remained implicit, the arts were reported as a 'black box' conceptually and practically (no knowledge of the internal workings).

Other limitations to the reporting of the arts elements of the interventions included few detailed analyses of who was involved in the activities, how they were invited, and which parts of the communities may have been excluded for cultural, social, political, or economic reasons.

Finally, there was little consideration of how the arts activities aligned or not with broader community activities or norms, such as the provision of hospitality.

## To what extent were the interventions sustainable?

Approximately half of the studies did not record long-term outcomes which makes it difficult to judge their sustainability. There are of course issues with measuring effectiveness long-term, such as leakage of intervention into control groups.[45 51] Key to sustainability is whether change can be maintained after the withdrawal of the research team.[52]

A number of the studies claimed that the engagement of local arts meant that the studies would be more sustainable. The reasons given were that the arts interventions promoted willingness to engage[53] or included celebration as a motivation for continuing.[51] Where interventions drew on local resources, this was often considered to make them more sustainable or low cost.[40 42 43 48 51 54] Others argued that the use of arts would be better for promoting retention/memory of the intervention messages[51 55] and that arts-based interventions help to create a positive attitude and contribute to people being more friendly to each other to promote sustainability.[40] Positive motivational drivers particularly embedded in arts-based messaging was seen to be more effective than negative drivers such as issuing defecation fines.[39]

Other themes on claims for sustainability were that the diverse and intensive nature of activities over time promoted sustainability,[39] and that interventions were more realistic to implement when they were embedded in local infrastructure and resources so that people can access the intervention.[44 45] It was, however, also noted that ensuring that those delivering the intervention are not distracted from their other duties when drawing on local infrastructure was also important, as well as ensuring appropriate additional incentivisation for involvement.[45 49] Local infrastructure issues, such as a lack of electricity or equipment, made the delivery of materials such as videos challenging and affected sustainability.[49] A final issue in sustainability was whether other members of the family and community (beyond the mother) were brought in to engage with the intervention[41] because broader family support was reported to bring longer-term sustainability of messaging.

## To what extent were interventions transferable?

Systematic participatory approaches simplify the process of contextual adaptation[53] enabling the process of developing and delivering the intervention to be replicated (even if the content is different). Interventions that use existing staff, for example, teachers or community health workers,[38 47] to deliver the intervention find that this promotes transferability into contexts where the health or education systems[54] or infrastructure[44 45] are similar and these professionals or paraprofessionals receive broadly similar training. However, there is always a risk that staff groups in different contexts may not buy into the

intervention. In addition, varying relationships between governments and NGOs were a limitation to transferability in some contexts.[52]

## To what extent were interventions scalable?

Small-scale interventions can limit the potential for scalability[39] particularly where they are reliant on individual motivation and momentum. Broad campaigns that capture large audiences quickly are more likely to be scalable, but the specificity of messaging can be challenging,[56] which may limit the effectiveness or traction of the messages. Elements of group interaction also maximise the efficient use of resources,[53] making the intervention more scalable, as well as having the potential to enhance community ties. A further point raised on scalability of arts-based interventions was when tried and tested health intervention materials were used alongside arts-based approaches leading to improved buy in for scaling.[57]

Major limitations around scalability within LMICs are cost and resource, with many of these interventions seeking to minimise the resource burden on government or communities.[40 41 43 47 52 54 55] Examples of this are using local storytelling[58] or theatre[55 57] which are cheaper than the production costs of video making.[49] Some costs can be prohibitive for scaling for example, TV ads[52] and cameras on smartphones.[44] For this reason, NGO support and finances can be critical requirements for scalability.[42 46 47 54 55 59]

## DISCUSSION
### Summary of findings
We identified three main programme theories, suggesting the importance of (i) participatory design based on valuing different sources of expertise; (ii) tailoring of the intervention to the local political and cultural context, and (iii) community participation in arts-based approaches. We found mixed evidence on sustainability of interventions, but that those embedded in local infrastructure and culture were more likely to be sustainable. We found that transferability of the interventions to other contexts, depended on the social and political organisation of health and child development activities (such as schools and community health workers). We found that the scalability of the community-arts interventions was variable, depending on the scale and levels of integration of the arts approaches within wider programmes of health and child development work as well as the availability of financial resources.

### Contribution to the literature (empirical)
Our findings on community-based arts interventions targeted at improving health and development outcomes for 0–5 years old in LMICs have revealed a number of gaps in current knowledge and understanding of what works for whom in which contexts.

The literature on interventions currently lacks input and leadership from academics working in the arts,

humanities and the social sciences. The lack of inclusion of arts scholars means there is a lack of theory integrated into discussions on how the arts were expected to support the interventions delivered, and it is unclear what was actually done in practice with local artists. Arts scholarship does engage with some of these issues[31] but not always in an interdisciplinary way where it is linked to outcomes of interventions (hence not meeting our review inclusion criteria). The lack of social science input likely leads to a dominant focus of the literature on outcomes as opposed to the processes used to achieve these. While there are literatures that suggest that the arts have the potential to break down power dynamics and provide communities a sense of ownership of the intervention with audiences owning the art or the performance,[60–62] these critical approaches were not used to interpret the findings for these interventions. This means that opportunities to learn about what works for whom and in which contexts were missed.

The lack of local lead authors of studies and insufficient studies from some regions including West and Southern Africa and South America means that contextual understanding is often lacking in the literature reviewed and this gap is particularly big for the regions with no pre-existing evidence.

The majority of the studies included in this review used a model of coproduction that involved bringing in communities to adapt existing interventions at a later stage of the development process as opposed to including communities from conceptualisation. The minority of studies that did work with communities throughout the process to coproduce were able to document benefits including time to build trust, community capacity, empowering local leadership and developing a true collaboration. These factors were all relevant for intervention sustainability because they may empower the community to take the intervention forward after the study ends. Overall, these elements were poorly reported, although this may reflect the focus and limited words counts in target journals.

Studies that entered communities early to codesign were better placed to draw on existing health systems including community health workers, social infrastructures such as fathers' or mothers' groups, and community facilities such as community spaces. Using existing infrastructures saves investment in time and resources to build and grow social systems and staffing infrastructures to support these types of intervention and supports sustainability. Studies that did use existing structures reported the importance of the level of adequate functioning of these systems to positively influence intervention outcomes. Ongoing training of workers or volunteers in the system was also reported to be important to maintain the success of intervention messaging and to support intervention sustainability beyond the study. This review also identified studies that did not report interaction with existing community structures such as health systems which was a missed opportunity for joined up thinking and promoting intervention sustainability.

## Contribution (theoretical)

The inter-relatedness and interdependence of the three themes (coproduction, local adaptation and arts-based delivery) underpins the theoretical contribution of this review. While coproduction and local adaptation offer two elements of 'scaffolding' that are necessary but not always sufficient for change, our findings suggest that the arts-based delivery approach can act as a bridge for overcoming some of the contextual and political challenges of delivering nurturing care interventions in LMICs. The use of realist methods for synthesising the literature allowed us to explore multiple and intersecting mechanisms of action, and to explore those mechanisms within different contexts. It allowed us to see that the success of the arts elements is dependent on effective collaboration and adapting to context, and to see that the arts can enhance and enact the benefits of collaborative working and careful local adaptation processes. Nevertheless, the lack of explicit reporting in papers on the mechanisms through which the arts work to enact change means that we are left with limits to our knowledge about what works, for whom, and in what circumstances. For instance we have learnt little about audience involvement in the arts; cultural embeddedness of the arts; how the arts works most effectively with the public health services; how the arts messages can be effectively shaped by those who know about the health and development elements; the role of knowledge and background in the health topic of the artists; inclusion or inclusion of certain population groups through the arts; the role of the arts venues, and how the arts interact with social norms. All of these would be important to gain understanding of the role of the arts in facilitating nurturing care.

There is evidence to suggest the importance of working toward synergy between partners[11] in both goals and processes,[63] and of including local people, community leaders, public health and education professionals, policymakers, and researchers, as well as funders of nurturing care interventions, in order to achieve more than each could alone.[64] Our findings supported this through stressing the value of regular formal[53] or informal feedback opportunities[40] and regular participatory meetings to support sustainability.[47] This builds trust[15 65] and transparency.[66 67]

When collaborations also acknowledge that interventions must be tailored to context in both the design and implementation phases, more can be achieved. Complementing or integrating with existing educational and health infrastructures and human resources was identified as being key, as was an understanding of cultural and social norms[39 42 49] for sustainable change.

However, a key risk to effective coproduction and tailoring to local context is the dysregulating effect of power imbalances. Enabling empowerment of those with less historical or cultural power in collaborations is a vital element of success.[16] The review has highlighted how challenging it is for collaborations to achieve high levels of balance and equity in contributions, and how rare it

is for these issues to be explicitly or critically reported. Our analysis can make an additional theoretical contribution here by identifying the potential for arts-based approaches to act as a bridge for some of these challenges of power imbalances through practical and political pathways.

First, *practically,* our findings suggest that engagement with community arts increases the potential effectiveness of the intervention in achieving (shared) goals of improved health and child development. Both the arts and community aspects support engagement in a shared experience of performance and through the congruence of local arts with local culture. Our findings suggest that there are two elements to this: increasing motivation to engage, and shifting long standing social norms. Positive motivation is engendered though things like a festival-like atmosphere, diversity of art forms, and the focus on positive impacts of change. The social dimensions of motivation may be particularly encouraged by communal experiences which contribute to group binding.[22 23] The use of local arts, local performers and organisers[55 57] and even local stories[42] increases the sense of ownership.[40] Interventions that are able to acknowledge and then shift long-established social norms from within are much more likely to be sustainable, rather than introducing external ideas. While the studies in the review rarely explicitly commented on social norms, our analysis was able to identify possible elements of the individual, social, material and institutional domains[68] in which social norms operate. The active engagement and the shared memories of an event or experience were considered to promote discussion of intervention materials, improve its retention individually and, importantly, its diffusion through social networks, reaching beyond intervention participants.[51 55]

Second, *politically,* our findings suggest that these sorts of community arts-based delivery approaches, when combined with coproduction and local adaptation, have the potential to disrupt some of the neo-colonial tendencies in global health research and practice. This involves shifting the balance of power away from Western biomedical models, towards communities and their historical and cultural modes of artistic expression and cultural re/production. Culturally embedded arts as opposed to non-culturally embedded arts have the most potential to achieve this outcome. Even the frequent delivery of these interventions in local languages suggests they are at least partially outwith the surveillance of global health researchers. While in some cases attempts to address power imbalances were evident, such as minority outreach and platforming during performances[58]; the targeting of women's empowerment through agricultural work and education[45]; targeting father and grandmother involvement in nurturing care practices,[41 48] or setting up partnership agreements,[69] in many cases this was not addressed explicitly in the papers.

### Strengths and limitations of the review

The review only captures published peer-reviewed journal articles. This likely results in a positive results publication bias and a lack of reports on arts-based interventions (grey literature). While we did not explicitly exclude French and Spanish articles (we had the language capacity in the team), we conducted the searches in English only. Despite searching a wide range of databases, we have identified that the papers that have been included are focused on health sciences rather than the arts/social science parts of the research. Many arts scholars publish in monographs or edited collections which are not always effectively captured through database searches.

### Recommendations for further research

There is a significant gap in the international literature, for studies that are led by researchers from LMICs and/or by those in the arts, humanities and social sciences. Bringing these voices into the research would enable better integration of the theoretical underpinnings regarding the potential success of arts-based intervention processes in nurturing care interventions. It would also facilitate better understanding of intervention processes and cultural context of these. Specific steps could be considered to ensure this can happen, such as leadership from LMICs, devolving funding decisions, and supporting and funding projects led from the arts, humanities and social sciences as well as the health and clinical sciences. Methods for ongoing reflective evaluation of the quality and equity in collaborations would be useful. Contextual issues may be relevant, including sources of funding being weighted toward health funders and the motivation for academics to publish in high-impact journals, which tend to be more STEMM-focused. Finally, reporting in journals of interventions with arts should include better documentation of their learning regarding the process of intervention delivery and the power dynamics involved in this.

### Recommendations for policy and practice

Nurturing care interventions are complex involving the health system, early education providers, social workers, families, extended families and community leadership structures. As we have shown in this review there are attempts taking place to incorporate the practical and political power of the arts into intervention design to empower communities to own nurturing care interventions, to facilitate message sharing and to embed intervention messaging into the culture of communities. Our findings suggest that these types of intervention have the potential for greater success in being contextually relevant when they include local knowledge including artists, are evaluated by teams that include those with experience in the arts, include collaborative work between artists and public health experts to adapt and deliver the intervention, and where local researchers take a lead or active role in the intervention delivery, evaluation and reporting.

There is also evidence in this review that where arts-based interventions embed into existing infrastructures (eg, health services or education) and there is a clear plan for ongoing resource for training of those delivering the intervention that sustainability is likely. To better understand what works, for whom, and in which contexts when designing nurturing care arts-based interventions, there is a need for studies to better document their learning regarding the process of intervention delivery and the power dynamics involved in this.

## CONCLUSIONS

The findings of this realist review contribute to our understanding of the potential impact of coproduced, arts interventions for nurturing care in LMICs. Using arts approaches has the potential to act as a bridge for some of the challenges of power imbalances between researchers, health and education professionals, and communities, that tailoring interventions to context and participatory design could not achieve alone. Interdisciplinary investigations integrating insights from arts scholarship and from the public health fields could support future research.

**Acknowledgements** In addition to our core research team, we would like to thank Rachel Posaner for her work developing and executing the search strategies for this paper. We would also like to thank the wider MaaCiwara team for their support.

**Contributors** NKG: securing funding; study design; data collection, extraction and analysis; manuscript drafting; manuscript development and review, guarantor. KA: study design; data collection, extraction and analysis; manuscript drafting; manuscript development and review. NHD: data collection, extraction and analysis; manuscript development and review. SM-H: securing funding; manuscript development and review. EA: manuscript development and review. CSS: securing funding; manuscript development and review. OT: securing funding; manuscript development and review. MW: manuscript development and review. PG: securing funding; study design; data collection, extraction and analysis; manuscript drafting; manuscript development and review.

**Funding** This research was funded by Medical Research Council (MRC), UK Research and Innovation (UKRI) Global Challenges Research Fund (GCRF) MR/T030011/1. The funder of this study has no role in the design, conduct, collection of data, analysis or writing of outputs.

**Competing interests** None declared.

**Patient and public involvement** Patients and/or the public were not involved in the design, or conduct, or reporting, or dissemination plans of this research.

**Patient consent for publication** Not applicable.

**Ethics approval** Not applicable.

**Provenance and peer review** Not commissioned; externally peer reviewed.

**Data availability statement** No data are available. All papers included in the review are available on the journal websites.

**ORCID iDs**
Nicola Kay Gale http://orcid.org/0000-0001-5295-8841
Cheick Sidya Sidibé http://orcid.org/0000-0002-7101-5408

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
