## [Reviewer comments · BMJ Open]

This paper was submitted to a another journal from BMJ but declined for publication following peer review. The authors addressed the reviewers' comments and submitted the revised paper to BMJ Open. The paper was subsequently accepted for publication at BMJ Open.

ARTICLE DETAILS

TITLE (PROVISIONAL)	Co-produced, arts interventions for nurturing care (0-5 years) in Low- and Middle-Income Countries (LMICs): a realist review.
AUTHORS	Gale, Nicola; Ahmed, Kalim; Diarra, Niélé; Manaseki-Holland, Semira; Asamane, Evans; Sidibé, Cheick; Touré, Ousmane; Wilson, Michael; Griffiths, Paula

VERSION 1 – REVIEW

REVIEWER	Christian, Aaron University of Ghana, Regional Institute For Population Studies
REVIEW RETURNED	20-Jan-2024

GENERAL COMMENTS	1. The justification of the study can be strengthened a bit in the Introduction.2. Indicators for which authors consider an intervention to have worked should clearly be stated or explained.3. Authors can provide a sentence or two why the choice of "realist Review"
---

REVIEWER	Barnish, Maxwell University of Exeter, Exeter Medical School
REVIEW RETURNED	22-Jan-2024

GENERAL COMMENTS	This work is innovative, of interest, and generally of a high standard. I do have some suggestions to improve the presentation of the work for publication. Firstly, there are a number of ways in which the manuscript does not conform to the BMJ Open instructions for authors. While these issues are numerous, they are readily rectifiable. 1. Please present a structured abstract according to BMJ Open instructions for authors.2. Please use unnumbered headings and structure the headings according to recent articles in this journal.3. Please use the BMJ Open referencing style both in-text and in the bibliography.4. Please ensure all articles are cited clearly – please remove the 'anonymous reference'.5. Please present a broad introduction before moving to the rationale and aims for your work. This would entail moving lines 40-48 to after line 125.
--

6. Please provide a Declarations section at the end of the manuscript according to BMJ Open instructions for authors.

7. Please provide a statement of the reporting guidelines used to inform this work at the start of the methods section (it would appear this is RAMESES) and provide a completed reporting checklist.

8. Please ensure that all tables are numbered correctly and are cited in the text. Table 3 does not have a numbered header.

9. Please move lengthy tables to the supplementary information, only keeping concise tables in the main paper.

10. Please ensure that text is not cut off, e.g. line 52 on page 56.

11. Please list the word count on the title page – it should be no more than 4000 words for the main body of the article (excluding title page, abstract, declarations and references).

12. Please include the authors' initials when you say who performed which tasks, such as data extraction.

Secondly, I have a few academic suggestions:

1. Please make minor amendments to the flow of information in the introduction, to avoid having two brief mentions of the arts (lines 66-71; line 73) that are not supported by citations, before you introduce the arts thoroughly in line 91 and following.
2. Please provide a reference for your definition of the Applied Arts (line 91).
3. It is not accurate to say that "Systematic reviews of arts-based interventions are lacking" (line 108). I've published two in the last few years (Barnish MS, Barran SM. A systematic review of active group-based dance, singing, music therapy and theatrical interventions for quality of life, functional communication, speech, motor function and cognitive status in people with Parkinson's disease. *BMC Neurology* 2020; 20: 371; Barnish MS, Nelson-Horne RV. Group-based active artistic interventions for adults with primary anxiety and depression: a systematic review. *BMJ Open* 2023; 13: e069310), and there could be others. You need to mention the broader systematic review evidence for the arts in health in different adult populations, before narrowing it down to your specific area of interest – children aged 0-5.
4. Similarly, your statement "Arts scholarship tends to focus less on evaluating interventions per se, and more on the artistry or process of artistic performance or creativity, or the role of artistic practice in interrogating concepts or cultures" may be accurate for some scholars who come from an arts background, but certainly not all, and is certainly not accurate for the many scholars from an epidemiology or community background who are interested in the arts and health. Most work in this field is in health journals not artistic journals.
5. It would be useful, before you move onto discussing the arts, to provide more framing about the health context in LMICs. My work in this area (Barnish MS, Tan SY, Taeihagh A, et al. Linking political exposures to child and maternal health outcomes: a realist review. *BMC Public Health* 2021; 21: 127; Barnish MS, Tan SY, Robinson S, et al. A realist synthesis to develop an explanatory model of how policy instruments impact child and maternal health outcomes *Soc Sci Med* 2023; 339: 116402) could be useful to cite, or so could work by others.
6. In various places, the expression in the article is long-winded and the argument slow to build. This affects the readability of the work and likely contributes to word count challenges.
7. Please ensure that the methods section reports what you did and the results section reports what you found. Currently, the

	methods section includes search results, which should be moved to the start of the results section (as per RAMESES guidelines). 8. Risk of bias is not typically done for realist reviews, however it would be worthwhile to add a sentence to the methods saying this, as readers of a general medical journal may be more accustomed to systematic reviews. 9. Please provide additional clarity regarding the methods for realist synthesis. In particular, the different stages of theory generation and adjudication, how many people were involved, was it consensus based, and was any translation involved? 10. Please provide a PRISMA flow chart. While it was designed for systematic reviews, the flow chart itself is very useful for realist reviews to see how many studies dropped out where. 11. Please provide as supplementary files a list of included studies and a list of studies excluded at full-text screening, grouped by primary reason for exclusion. 12. I like the section on authorship location and discipline – it would be useful to cite the relevant articles to show which studies were based where, so people can cross-check this with the results and help their interpretation of the richness of the findings. 13. Please provide as a supplementary file a table of the key results for each study. 14. Please provide an overall conclusion at the end of the article. Most of these suggestions are a fairly quick fix as well. Please also do a sweep of the article for English language and spelling issues, such as one mention of 'UNCEF' rather than 'UNICEF'.
--	--

VERSION 1 – AUTHOR RESPONSE

Reviewer 1	
1. The justification of the study can be strengthened a bit in the Introduction.	The introduction has been expanded to include information on the significance/justification of the study (moved to after the background, as per journal guidance).
2. Indicators for which authors consider an intervention to have worked should clearly be stated or explained.	Given the realist approach taken, and given the heterogeneity of the interventions, their desired outcomes and the methods used to evaluate them, it was not possible to have a single standard of effectiveness. Intermediary outcomes were identified in some papers but reporting was often incomplete. Further clarification of this has been provided in the methods section.
3. Authors can provide a sentence or two why the choice of "realist Review"	This is in a section at the end of the edited 'background' section that covers this. We have also added further information at the beginning of the Methods section.
Reviewer 2	

This work is innovative, of interest, and generally of a high standard. I do have some suggestions to improve the presentation of the work for publication.	Many thanks and for the useful comments which we have addressed below.
Firstly, there are a number of ways in which the manuscript does not conform to the BMJ Open instructions for authors. While these issues are numerous, they are readily rectifiable.  1. Please present a structured abstract according to BMJ Open instructions for authors. 2. Please use unnumbered headings and structure the headings according to recent articles in this journal. 3. Please use the BMJ Open referencing style both in-text and in the bibliography. 4. Please ensure all articles are cited clearly – please remove the ‘anonymous reference’. 5. Please present a broad introduction before moving to the rationale and aims for your work. This would entail moving lines 40-48 to after line 125. 6. Please provide a Declarations section at the end of the manuscript according to BMJ Open instructions for authors. 7. Please provide a statement of the reporting guidelines used to inform this work at the start of the methods section (it would appear this is RAMESES) and provide a completed reporting checklist. 8. Please ensure that all tables are numbered correctly and are cited in the text. Table 3 does not have a numbered header. 9. Please move lengthy tables to the supplementary information, only keeping concise tables in the main paper. 10. Please ensure that text is not cut off, e.g. line 52 on page 56. 11. Please list the word count on the title page – it should be no more than 4000 words for the main body of the article (excluding title page, abstract, declarations and references). 	We have gone through each of these and corrected (see also editor’s comments above). Referencing has been updated to BMJ Open referencing style. Word count has been reduced as far as possible, noting also that we’ve had to include additional things in response to the reviewers.

12. Please include the authors' initials when you say who performed which tasks, such as data extraction.	
Secondly, I have a few academic suggestions: 1. Please make minor amendments to the flow of information in the introduction, to avoid having two brief mentions of the arts (lines 66-71; line 73) that are not supported by citations, before you introduce the arts thoroughly in line 91 and following.	Thanks for this helpful observation about the flow of information in the background section.
2. Please provide a reference for your definition of the Applied Arts (line 91).	Included references.
3. It is not accurate to say that "Systematic reviews of arts-based interventions are lacking" (line 108). I've published two in the last few years (Barnish MS, Barran SM. A systematic review of active group-based dance, singing, music therapy and theatrical interventions for quality of life, functional communication, speech, motor function and cognitive status in people with Parkinson's disease. BMC Neurology 2020; 20: 371; Barnish MS, Nelson-Horne RV. Group-based active artistic interventions for adults with primary anxiety and depression: a systematic review. BMJ Open 2023; 13: e069310), and there could be others. You need to mention the broader systematic review evidence for the arts in health in different adult populations, before narrowing it down to your specific area of interest – children aged 0-5.	Thanks for picking up this statement which was indeed too sweeping. We've edited for clarity and context.
4. Similarly, your statement "Arts scholarship tends to focus less on evaluating interventions per se, and more on the artistry or process of artistic performance or creativity, or the role of artistic practice in interrogating concepts or cultures" may be accurate for some scholars who come from an arts background, but certainly not all, and is certainly not accurate for the many scholars from an epidemiology or community background who are interested in the arts and health. Most work in this field is in health journals not artistic journals.	We agreed that there is a lot of interdisciplinary work (including our own), but in this case we are referring to single discipline work and this statement was informed by our arts-based co-author who is an expert in the field. We are referring to scholarship from arts disciplines. We are not suggesting that this is true of all arts scholarship, but rather that there is a tendency towards this approach, highly influenced by the expectations of journals. We have amended the text to clarify and to distinguish this from other important interdisciplinary work that draws on the arts and is published in health journals.
5. It would be useful, before you move onto discussing the arts, to provide more framing about the health context in LMICs. My work in this area (Barnish MS, Tan SY, Taeihagh A, et	Thanks for these useful references and suggestion, which have been incorporated.

al. Linking political exposures to child and maternal health outcomes: a realist review. BMC Public Health 2021; 21: 127; Barnish MS, Tan SY, Robinson S, et al. A realist synthesis to develop an explanatory model of how policy instruments impact child and maternal health outcomes Soc Sci Med 2023; 339: 116402) could be useful to cite, or so could work by others.	
6. In various places, the expression in the article is long-winded and the argument slow to build. This affects the readability of the work and likely contributes to word count challenges.	The paper has been re-reviewed post revisions for readability.
7. Please ensure that the methods section reports what you did and the results section reports what you found. Currently, the methods section includes search results, which should be moved to the start of the results section (as per RAMESES guidelines).	This was an oversight and has been amended and reporting is fully in line with RAMESES now.
8. Risk of bias is not typically done for realist reviews, however it would be worthwhile to add a sentence to the methods saying this, as readers of a general medical journal may be more accustomed to systematic reviews.	Noted, and added.
9. Please provide additional clarity regarding the methods for realist synthesis. In particular, the different stages of theory generation and adjudication, how many people were involved, was it consensus based, and was any translation involved?	Further clarification is provided. We drew on the Sayer methods of identifying demi-regularities most heavily during this process.
10. Please provide a PRISMA flow chart. While it was designed for systematic reviews, the flow chart itself is very useful for realist reviews to see how many studies dropped out where.	Added as supplementary file.
11. Please provide as supplementary files a list of included studies and a list of studies excluded at full-text screening, grouped by primary reason for exclusion.	This information is summarised in Table 2 and additional detail provided in the supplementary file. RAMESES guidelines do not require the full list of excluded papers but I will defer to the Editor for guidance on this.
12. I like the section on authorship location and discipline – it would be useful to cite the relevant articles to show which studies were based where, so people can cross-check this with the results and help their interpretation of the richness of the findings.	The information is in Table 3. In addition, we have added the citations directly in the text.

13. Please provide as a supplementary file a table of the key results for each study.	This information has been added to Table 3.
14. Please provide an overall conclusion at the end of the article	A conclusion has been added.
Please also do a sweep of the article for English language and spelling issues, such as one mention of 'UNCEF' rather than 'UNICEF'	This has been done.

VERSION 2 – REVIEW

REVIEWER	Barnish, Maxwell University of Exeter, Exeter Medical School
REVIEW RETURNED	22-Mar-2024

GENERAL COMMENTS	I consider that the authors have responded well to all points raised by the reviewers and made necessary amendments. I would prefer a full list of the studies excluded at full text stage to be included as an appendix, but I too will defer to the editor's judgement on this.
---